# Does the Use of Local Antibiotics Affect Clinical Outcome of Patients with Fracture-Related Infection?

**DOI:** 10.3390/antibiotics11101330

**Published:** 2022-09-29

**Authors:** Jonathan Sliepen, Ruth A. Corrigan, Maria Dudareva, Marjan Wouthuyzen-Bakker, Rob J. Rentenaar, Bridget L. Atkins, Geertje A. M. Govaert, Martin A. McNally, Frank F. A. IJpma

**Affiliations:** 1Department of Trauma Surgery, University Medical Center Groningen, University of Groningen, 9712 CP Groningen, The Netherlands; 2Department of Infectious Diseases, Nuffield Orthopaedic Centre, Oxford University Hospitals, Oxford OX3 9DU, UK; 3Department of Medical Microbiology and Infection Prevention, University Medical Center Groningen, University of Groningen, 9712 CP Groningen, The Netherlands; 4Department of Medical Microbiology, University Medical Center Utrecht, 3584 CX Utrecht, The Netherlands; 5Department of Trauma Surgery, University Medical Center Utrecht, 3584 CX Utrecht, The Netherlands; 6The Bone Infection Unit, Nuffield Orthopaedic Centre, Oxford University Hospitals, Oxford OX3 9DU, UK

**Keywords:** fracture-related infection, fracture, infection, antibiotic-loaded carriers, local antibiotics

## Abstract

This international, multi-center study evaluated the effect of antibiotic-loaded carriers (ALCs) on outcome in patients with a fracture-related infection (FRI) and evaluated whether bacterial resistance to the implanted antibiotics influences their efficacy. All patients who were retrospectively diagnosed with FRI according to the FRI consensus definition, between January 2015 and December 2019, and who underwent surgical treatment for FRI at any time point after injury, were considered for inclusion. Patients were followed-up for at least 12 months. The primary outcome was the recurrence rate of FRI at follow-up. Inverse probability for treatment weighting (IPTW) modeling and multivariable regression analyses were used to assess the relationship between the application of ALCs and recurrence rate of FRI at 12 months and 24 months. Overall, 429 patients with 433 FRIs were included. A total of 251 (58.0%) cases were treated with ALCs. Gentamicin was the most frequently used antibiotic (247/251). Recurrence of infection after surgery occurred in 25/251 (10%) patients who received ALCs and in 34/182 (18.7%) patients who did not (unadjusted hazard ratio (uHR): 0.48, 95% CI: [0.29–0.81]). Resistance of cultured microorganisms to the implanted antibiotic was not associated with a higher risk of recurrence of FRI (uHR: 0.75, 95% CI: [0.32–1.74]). The application of ALCs in treatment of FRI is likely to reduce the risk of recurrence of infection. The high antibiotic concentrations of ALCs eradicate most pathogens regardless of susceptibility test results.

## 1. Introduction

### 1.1. Background

Fracture-related infection (FRI) is a major complication that can occur after fracture care. FRI comes with substantial socioeconomic costs. The direct hospital-related healthcare costs can be up to eight times higher compared to non-infected fractures [1]. The cornerstones of FRI treatment are debridement of non-viable tissues, a thorough lavage, fracture stabilization, soft tissue coverage and systemic antimicrobial therapy [2,3,4,5]. Any bony defect that arises as a result of dead bone removal needs to be managed, as this cavity (the dead space) can be an ideal environment for haematoma formation and bacterial proliferation. Dead spaces can be managed in several ways, including the use of antibiotic-loaded carriers (ALCs). Various ALCs are available, all with their own biologic characteristics and antibiotic releasing profile [6,7,8,9]. All commonly used carriers release high concentrations of the applied antibiotic in the first days after surgery. A major benefit of ALCs compared to systemic antibiotic treatment, apart from their ability in reducing the actual cavity, is that much higher antibiotic concentrations can be achieved, even in poorly vascularized areas, without causing systemic side effects [10,11,12,13]. Using ALCs, concentrations up to 10–100 times higher than the minimum inhibitory concentration (MIC) of common clinically relevant bacteria can be achieved [9,14].

### 1.2. Rationale

The application of ALCs for the prevention and treatment of FRI has received much attention in recent decades and is suggested to improve outcome [15,16,17,18]. Numerous studies focusing on the application of ALCs show recurrence rates of infection ranging from 0 to 23.3% [18,19,20,21,22,23,24,25,26,27,28,29]. However, most of these studies did not use a standardized definition to diagnose infection and included patient populations with chronic osteomyelitis arising from both posttraumatic and haematogenous (or other) origins. Although the eradication rates of infection vary substantially between studies, several case series showed promising results for the use of ALCs [19,20,22,23,28]. Nonetheless, high-quality prospective studies or randomized controlled trials demonstrating the correct indication and/or supporting the benefit of ALCs are still lacking [3,25,30].

Ideally, the administration of antimicrobial agents is tailored to the resistance pattern of the causative microorganism [5]. However, in the majority of FRI cases, an ALC is implanted when the causative agent is still unknown. Another problem is that bacteria in biofilms can be difficult to detect and identify using conventional tissue culture techniques [31]. Biofilm-producing bacteria are more difficult to treat and biofilm production is associated with worse clinical outcome [32]. Additionally, some studies assume that recurrence of infection is more likely in cases where pathogens are resistant to the implanted antibiotic [19,33,34]. However, in vitro studies demonstrate antibiotic concentrations far above the minimum inhibitory concentration (MIC) and even above the minimum biofilm eradication concentration (MBEC) in the first days after application of local antimicrobials [35]. We hypothesized that this high level of antibiotics delivered locally in the first days after surgery might eradicate pathogens regardless of laboratory tested susceptibility. Clinical data on the relationship between in vitro bacterial resistance to these ALCs and the effect on treatment outcome of FRI are still lacking.

### 1.3. Research Questions

This study assessed the relationship between the application of ALCs and management of FRI. The following research questions were posed: (1) Does the use of ALCs affect infection recurrence rate, when implanted in or around infected fractures during surgical treatment of FRI? and (2) is bacterial resistance to the implanted antibiotic associated with a poorer outcome?

## 2. Results

### 2.1. Patient Population

A total of 429 patients with 433 FRIs were included. The median age was 50.6 (range: 17–84) years and 318 (73.4%) were males. The most frequently involved anatomical localizations were the tibia (50.6%) and femur (21.7%). Overall, *Staphylococcus aureus* was the most frequently cultured microorganism, followed by Gram-negative bacteria (GNB) and Coagulase-negative staphylococci (CoNS) (Figure 1). ALCs were used in 251/433 (58.0%) cases (Table 1). In the group treated with ALCs compared to those without ALCs, the median time to onset of infection was longer (125.0 weeks (P_25_–P_75_: 39.0–806.0) vs. 6.5 weeks (P_25_–P_75_: 3.0–38.0), *p* < 0.001) and there were more cases with a healed fracture (64.8% vs. 14.8%, *p* < 0.001). The surgical approach also differed between both groups. In the ALC group, surgery without the need for fracture fixation (37.7% vs. 6.0%, *p* < 0.001), exchange to external fixation (20.3% vs. 9.3%, *p* = 0.002) and removal of any fixation device (29.1% vs. 12.1%) were more frequently performed. In the group without ALCs, DAIR was more often performed (64.3% vs. 8.8%, *p* < 0.001). The pathogens cultured at initial FRI treatment compared to the pathogens cultured at time of recurrent FRI are displayed in Appendix A (Appendix A). Identical pathogens, isolated at the initial FRI treatment and at the time of recurrence, were found in 10/34 (29.4%) of cases in the non-ALC group and in 9/25 (36.0%) of cases in the ALC group. Culture-negative cases were more common in the ALC group (25.1% vs. 9.3%, *p* < 0.001), while polymicrobial infections were more common in the group without ALCs (48.6% vs. 26.2%, *p* < 0.001). A total of 59 (13.6%) patients had a recurrence of FRI at final follow-up (Table 2).

### 2.2. Type of Local Antibiotics

In 213 out of 251 (84.9%) FRI cases treated with ALCs, the antibiotic was delivered in a bioabsorbable antibiotic carrier (Table 2). Gentamicin is the most frequently used and widely studied antibiotics in ALCs. It is thermally stable and provides broad bactericidal coverage, which makes it a suitable agent for ALCs. It was the most commonly implanted antibiotic and was applied in 247/251 (98.4%) cases (Table 3). In 223 of those cases, gentamicin was the only implanted antibiotic. In the remaining 24 cases, gentamicin was mixed with vancomycin (8 cases), tobramycin (8 cases), clindamycin (3 cases) and vancomycin and colistin (1 case). Table 4 provides an overview of the used antibiotic carrier vehicles. The outcomes according to the use of, and susceptibility to, ALCs are depicted in the flowchart (Appendix A).

### 2.3. Application of Antibiotic-Loaded Carriers in Relation to Outcome

Recurrence of infection at all time points after surgery occurred in 25/251 (10%) of patients treated with an ALC and in 34/182 (18.7%) patients treated without an ALC (HR: 0.48, 95% CI: [0.29–0.81]) (Table 2). This is also depicted in Figure 2 by the higher survival rate in the Kaplan–Meier survivorship curve for patients treated with ALCs. The unadjusted analyses of the effect of ALCs in relation to outcome (Figure 3) showed a significantly reduced risk of recurrence of infection at 12 months (HR: 0.42, 95% CI: [0.2–0.83] and 24 months (HR: 0.39, 95% CI: [0.2–0.74] when ALCs were applied. When adjusting for the IPTW propensity scores the models showed a consistent reduction in risk of recurrence that did not reach statistical significance at 12 months (HR: 0.69, 95% CI: [0.24–1.96]) and 24 months (HR:0.55, 95% CI: [0.22–1.35]). Across the whole cohort, multivariable regression also showed a consistent reduction in risk of recurrence that did not reach statistical significance at 12 months (HR: 0.71, 95% CI: [0.26–1.92]) and 24 months (HR: 0.46, 95% CI: [0.18–1.2]) when ALCs were applied. Subgroup analysis of patients with a late/chronic infection (n = 293, 67.7%), which was defined as an infection with a time to onset > 10 weeks according to the Willenegger and Roth classification [36], showed a significant reduction in recurrence of infection at 24 months (HR: 0.21, 95% CI: [0.06–0.96]) when an ALC was applied.

### 2.4. Bacterial Resistance to Local Antibiotics in Relation to Outcome

In 178/251 (70.9%) cases, the cultured microorganisms were susceptible to the implanted antibiotic. In 66 (26.3%) cases, the cultured microorganisms were resistant. Patients who were not treated with ALCs and seven patients with microorganisms with an unknown susceptibility to the applied ALCs were excluded from this analysis. Resistance of cultured microorganisms to the implanted antibiotic was not associated with a higher risk of recurrence of FRI (HR: 0.75, 95% CI: [0.32–1.74]) (Table 2). This is also depicted in Figure 4 by the overlapping lines of the Kaplan–Meier survivorship curve.

## 3. Discussion

This large multinational study, including patients with FRI at all time points after injury, showed that the use of an antibiotic-loaded carrier implanted in or around a fracture during the surgical treatment of FRI is likely to reduce the risk of recurrence of infection. Furthermore, bacterial resistance to the implanted antibiotics was found in a quarter of cases but did not result in poorer treatment outcomes. However, the use of ALCs, and the effect on outcome, was not uniformly distributed, as the IPTW analysis revealed. Other factors, such as the presence of an unhealed fracture, the use of DAIR or the nature of the infection (culture negative or polymicrobial) also had a significant influence. When these factors were included in the IPTW model, the confidence interval of the effect of ALCs was widened, suggesting that these factors also contribute to outcome. Nevertheless, in all analyses (Figure 3), the hazard ratio remained in favour of the use of ALCs, making it likely that ALCs did contribute to improved outcome, even in those clinical scenarios with a higher recurrence risk (i.e., unhealed fracture).

ALCs were used more frequently in cases with a longer time from injury, although the range of duration of infection was large in both groups (ALC group: 1–3432 weeks, non-ALC group: 0–1560 weeks). Longstanding FRIs tend to have had multiple previous operations, have poor soft tissues, occur in older, more co-morbid patients and may be regarded as more difficult to treat [37,38]. The data from our study showed that there was no increased recurrence rate in these longer duration FRIs when ALCs were used. In the subgroup of patients with a late/chronic FRI the use of ALCs significantly reduced the risk of recurrence of FRI (HR: 0.21, 95% CI: [0.06–0.96]) at 24 months follow-up. ALCs may therefore be best indicated in this type of FRI, although further investigation is needed. However, in a separate analysis from our study group, time from injury was not shown to be an independent factor in the outcome of surgical treatment of FRI [39], so the relationship between time and outcome may be more complex.

Bacterial resistance to the implanted antibiotics was observed in a quarter of patients treated with ALCs. However, the presence of bacterial resistance to the implanted antibiotic did not result in a poorer treatment outcome, which supports our hypothesis that high local concentrations may help to eradicate pathogens regardless of laboratory susceptibility. Alternative reasons for this effect could be that systemic antimicrobial therapy acted synergistically with ALCs and was more effective in eradicating infections resistant to the implanted antibiotic. Furthermore, it is possible that ALC was acting as broad-spectrum antimicrobial prophylaxis, preventing re-infection with a new organism. This is supported by studies showing that orthopaedic infection recurrence is commonly with different organisms to the original pathogens identified at index surgery [40], and by the effectiveness of ALCs in primary prophylaxis of FRI [15].

### 3.1. Application of Local Antibiotics Related to Outcome

Previous studies on the use of ALCs in treatment of chronic osteomyelitis with various origins reported variable results. Ferguson et al. described the use of biodegradable antibiotic-loaded calcium sulphate carrier containing tobramycin in a large cohort of 195 patients suffering chronic osteomyelitis (including 110 FRIs) and found a recurrence rate of 9.2% (mean follow-up of 3.7 years), which is in line with our results [23]. McNally et al. reported on the application of an absorbable, gentamicin-loaded, calcium sulphate/hydroxyapatite ceramic and found a recurrence rate of only 6% in 100 patients treated with a single-stage operation for chronic osteomyelitis (including 71 FRIs) [19]. Zhou et al. found a recurrence rate of 6% in 100 patients treated for posttraumatic osteomyelitis with an antibiotic-loaded calcium sulfate vehicle [41]. Furthermore, a review by Pesch et al. evaluated the treatment of FRI/osteomyelitis cases with a ceramic bone substitute in eight studies, including their own patient cohort and found a mean recurrence rate of 8.3% (range: 0–14.3%) [42]. Additionally, in a systematic review by Pincher et al. of single-stage revisions in osteomyelitis cases, the recurrence rate ranged from 2.6% to 20.7% in cases treated with an ALC [43]. Other smaller studies (number of patients treated with ALCs ranged from 20 to 30) demonstrated recurrence rates of 20% to 23.3% [27,44,45].

This is the largest comparative clinical study on ALCs, including patients at all time points after injury managed with a variety of appropriate surgical treatment strategies and various types of ALCs. It adds to the existing literature of smaller studies and those focusing on one type of antibiotic carrier. Based on the adjusted risk reduction on recurrence of infection in the entire cohort (HR: 0.55; 95% CI: [0.22–1.35] at 24 months follow-up) and the adjusted risk reduction in patients with a late/chronic FRI with a time to onset of >10 weeks (HR: 0.21, 95% CI: [0.06–0.96] at 24 months follow-up), the use of ALCs in treatment of FRI is likely to reduce the risk of recurrent FRI. The benefit of ALCs in the treatment of FRI also depends on good debridement, systemic antimicrobials, stabilization and soft-tissue coverage. Furthermore, the application of ALCs is considered safe [46] and systemic side effects have been rarely reported in the literature, yet, they should not be neglected [47]. However, further research is necessary to set the correct indication for the application of ALCs and on development of optimal carriers.

### 3.2. Susceptibility to Local Antibiotics Related to Outcome

Large clinical studies on the effect of bacterial resistance to ALCs are lacking. In our study, the treatment outcome for FRI patients with causative microorganisms resistant to the implanted antibiotic was not worse than for those patients with susceptible microorganisms. Two studies by McKee et al. investigating the effect of a tobramycin loaded bioabsorbable bone substitute reported that tobramycin is effective against most osteomyelitis causing species and the high concentrations of locally applied antibiotics may even eradicate organisms resistant to systemic doses [18,28]. McNally et al. found no substantial differences in recurrence of infection between patients with resistant organisms to ALCs (1/16, 6.25%) compared to those with organisms fully susceptible to ALCs (5/84, 5.95%; *p* = 0.958) in a cohort of 100 patients [19]. Ruppen et al. assessed in vitro activity of locally administered gentamicin, solely or as an adjunct to penicillin, against biofilm forming Group B Streptococci (GBS) [48]. They found that the high concentrations of antibiotics needed in bone to achieve activity against biofilm GBS could only be reached with local rather than systemic antibiotics. In line with the findings of Ruppen et al., we hypothesized that high levels of antimicrobials delivered locally could eradicate most pathogens, regardless of susceptibility. In our study, patients with microorganisms resistant to ALCs compared to those without resistant microorganisms did not have higher recurrence rates of FRI at follow-up, which can be considered a clinical substantiation of our hypothesis. Based on these results, physicians should be aware that the presence of bacterial resistance to ALCs may not have major consequences for prognosis, provided appropriate systemic antibiotic therapy is given [49].

This study does not distinguish between individual carrier materials or products. However, all cases were treated with antibiotics delivered in a commercially available carrier material, as can be seen in Table 4. These materials have evidence showing that they deliver high concentrations of antibiotic, with predictable elution curves over a period of several weeks. The results should not be extrapolated to the use of raw powder antibiotics inserted directly into the wound, without a carrier material.

### 3.3. Limitations

There are some limitations to this study that need to be addressed. Firstly, the retrospective nature of the study makes it prone to information and selection bias. To minimize the risk of information bias, data were cross-checked by multiple researchers and blinded. Data on ALCs were well documented; infection diagnosis and outcome were rigorously defined. Secondly, the ALC group and the non-ALC group differed in several respects. The difference in time to onset of infection, inevitably affected the number of healed fractures and thus the surgical approach, particularly the need for stabilization or the use of DAIR (selection bias). However, this was largely accounted for by the IPTW propensity score to make an estimation of likelihood that ALCs would have been used in a given case. Although the IPTW model and the multivariable models adjust for some potential confounding factors, in assessing the relationship between the application of ALCs and recurrence rate it was not possible to adjust for all confounding by indication completely. There are likely to be unmeasured factors such as infection severity that may have influenced surgeons’ decisions to retain or re-implant metalwork and therefore impact on whether or not an ALC was used. We based severity of infection on the BACH classification for osteomyelitis, but this has not been validated in FRI. However, it is likely that there remain unmeasured factors that could influence the infection severity (e.g., intracellular hiding bacteria). This study is not a randomized trial of ALCs in FRI, but the results suggest that such a study would be valuable.

## 4. Materials and Methods

### 4.1. Study Design

A multinational retrospective cohort study in which data of all patients with an FRI, occurring at all time points after initial fracture treatment, between January 2015 and December 2019 at the Bone Infection Unit in Oxford (United Kingdom), the University Medical Center Utrecht (The Netherlands) and University Medical Center Groningen (The Netherlands) were evaluated.

### 4.2. Study Population

All patients included in this study were retrospectively diagnosed with an FRI according to the FRI consensus criteria [50,51]. All patients were treated according to the ‘intention to treat’ principle, based on recommendations from a multidisciplinary team. This means that all patients were confirmed to have an infection and were treated as such. In all cases at least three surgically obtained deep tissue cultures were obtained. Patients with less than 12 months of follow-up after index surgical therapy, a pathological fracture or a fracture to the skull or spine were excluded. Electronic patient files were thoroughly reviewed and data on demographics, time to onset of symptoms, surgical procedure, culture results, bacterial susceptibility and use and details of the ALCs were gathered.

### 4.3. Surgical and Antibiotic Treatment

Surgical treatment consisted of deep tissue sampling (≥3 samples), debridement of dead or poorly vascularized tissue, adequate irrigation, stabilization when the fracture was not healed and soft tissue coverage [39].The decision to retain, replace or remove osteosynthesis materials depended on whether the fracture was healed or not. In case the fracture was not healed, and stability was required, either an external fixator was used or new internal fixation was performed in a (single) staged fashion to facilitate fracture healing. Generally, internal fixation was only chosen in cases where external fixation would be difficult and when the soft tissue was good or could be improved with soft tissue reconstruction. The indication whether or not to use local antibiotics was based on local hospital guidelines and/or surgeon’s preferences. Manufactured ALCs of various brands were used according to company instructions. This study solely focused on patients treated with local antibiotics delivered in a carrier vehicle. The application of raw antibiotic powder sprinkled in the wound as a local antibiotic was not part of this study. The systemic antibiotic treatment of patients in this cohort is described in more detail in a separate study [49].

### 4.4. Clinical Outcome

A team of trauma surgeons (GG, MMcN, FIJ) verified the surgical treatment and the clinical outcome in the patients’ records. A team of microbiologists/infectious disease specialists (RC, MWB, RR, BA) verified the appropriateness of the local and systemic antimicrobial therapy [52]. Both teams were blinded from each other’s findings. Primary outcome was the recurrence rate of infection at a minimum of one year after cessation of surgical treatment. Recurrence of infection, amputation of the affected limb due to infection and infection-related deaths were considered failure of treatment. Recurrence of infection was defined as the reappearance of any of the confirmatory signs according to the FRI consensus definition at follow-up [51,53].

### 4.5. Causative Pathogens and Local Antibiotics

Diagnosis of FRI was confirmed microbiologically when any phenotypically indistinguishable microorganisms were isolated from two or more surgically obtained deep tissue cultures, including sonication in case of removed implants [50,51]. When the diagnosis of FRI was based on non-microbiological criteria, virulent pathogens that are often the cause of FRI and were isolated from a single deep tissue culture were also considered as causative pathogens [5,53]. The term ‘virulent pathogens’ refers to pathogens with the ability to cause disease (infection) in the host and for this reason are likely to be clinically relevant. The following microorganisms were a priori defined as virulent pathogens: Gram-negative bacilli, *Staphylococcus aureus*, *Staphylococcus lugdunensis*, enterococci, beta-hemolytic streptococci, anginosus (milleri) group streptococci, *Streptococcus pneumoniae*, and Candida species [54]. The list of virulent pathogens is based on consensus opinion and the expertise of the microbiologists and infectious disease physicians. Other microorganisms with only a single positive culture were considered contaminants and were not considered further in this study. Data on the type of carrier vehicle (bioabsorbable or non-bioabsorbable) were noted. The type of antibiotic in or added to the carrier was gathered to evaluate the effect of bacterial resistance on outcome of patients treated with ALCs. To determine whether the locally applied antibiotics covered all cultured microorganisms, bacterial susceptibility was determined on laboratory testing of isolates, using the European Committee on Antimicrobial Susceptibility Testing (EUCAST) breakpoints. ‘ALCs covering all microorganisms’ was defined as all cultured microorganisms being susceptible to the locally applied antibiotics. Patients having a culture negative FRI were also considered as ‘ALCs covering all microorganisms’.

### 4.6. Statistical Analysis

For non-continuous variables, chi-square tests and Fisher-exact tests were performed. For continuous variables, independent-samples t-tests (reporting mean and standard deviation (SD)) were performed for normally distributed variables and Mann–Whitney U tests (reporting median and inter-quartile rage as 25th–75th percentile(P_25_–P_75_)) were performed for non-normally distributed continuous variables. Normality of continuous data was tested using Shapiro–Wilks test. To adjust for confounding factors, inverse probability for treatment weighting (IPTW) modeling and multivariable logistic regression analyses were used to evaluate the effect of ALCs on treatment outcome. All health-related baseline characteristics (sex, age, diabetes, smoking status, immunosuppressive medication, involved bone, bone healing status, presence of a sinus or pus, time to surgery and type of surgery) without missing data were used to calculate a IPTW propensity score. This propensity score balances the baseline patient characteristics in the ALC and non-ALC group by weighting each individual in the analysis by the inverse probability of receiving his/her actual treatment. It allows the contribution of other characteristics (such as time from injury, surgical strategy, etc.) to be quantified and to demonstrate how these impact on the effect of using ALC or not. A Cox proportional hazard model was used to evaluate outcome at final follow-up. To determine outcome at 12 and 24 months follow-up, two subsets were created including only patients who had at least 12 or 24 months of follow-up, respectively. All patients who were lost to follow-up, died or underwent amputation of the affected limb, were censored for further analysis. To evaluate outcome at 12 and 24 months follow-up IPTW modeling was performed as well as both unadjusted and adjusted logistic regression analysis. Results were presented using *p*-values or hazard ratios (HR) including a 95% confidence interval (95% CI). A significance level that was accepted for all tests was a *p* < 0.05 or a 95% CI not including one. Data were analyzed and visualized with IBM SPSS for Windows version 23.0 (IBM Corp., Armonk, NY, USA) and Rstudio for windows version 4.1.2. (RStudio Team (2020). RStudio: Integrated Development for R. RStudio, PBC, Boston, MA, USA).

## 5. Conclusions

The application of ALCs in treatment of FRI is likely to reduce the risk of recurrence of infection. The degree of this benefit may vary in different clinical scenarios. Large well-designed randomized controlled trials could provide a definitive answer on the potential effect of ALCs on outcomes in patients with FRI. Bacterial resistance to the implanted antibiotics was found in a quarter of cases, but this did not result in poorer treatment outcomes. Our clinical findings support the hypothesis that high local antibiotic concentrations eradicate most pathogens regardless of susceptibility. Moreover, the in vitro susceptibility test results of the causative organisms will not predict the therapeutic effect of ALCs, but remain instrumental in guiding systemic antibiotic treatment in FRI.

## Figures and Tables

**Figure 1 antibiotics-11-01330-f001:**
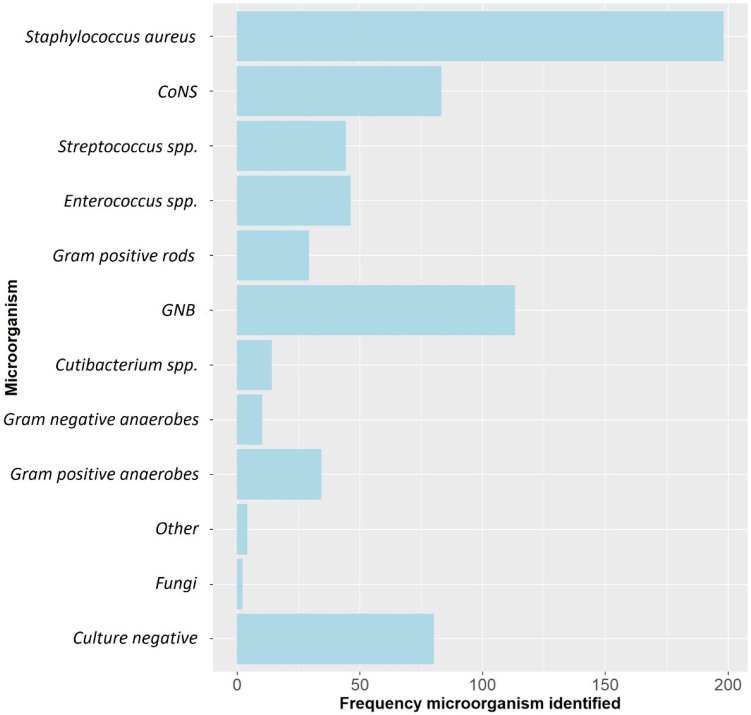
The microbiological epidemiology of this FRI cohort. CoNS: Coagulase-negative staphylococci, GNB: Gram-negative bacteria, spp: species.

**Figure 2 antibiotics-11-01330-f002:**
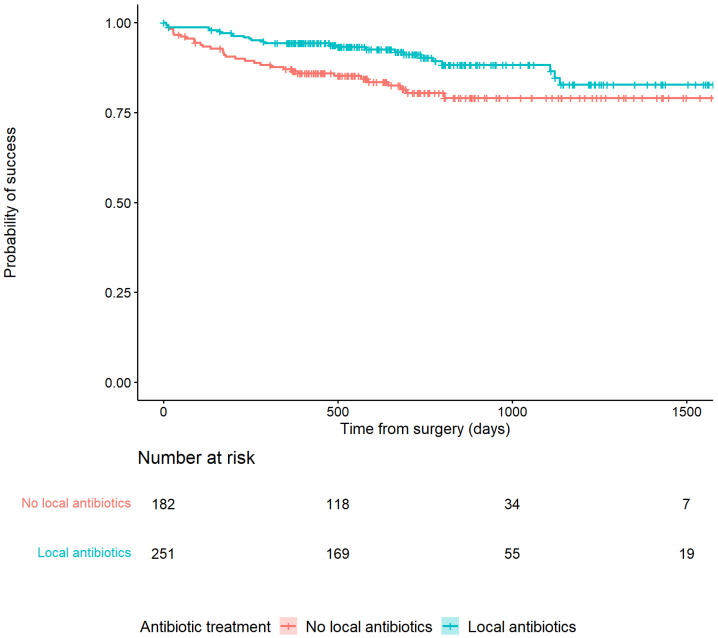
Kaplan–Meier Survivorship curve demonstrating the relationship between patients treated with or without antibiotic-loaded carriers and recurrence of infection.

**Figure 3 antibiotics-11-01330-f003:**
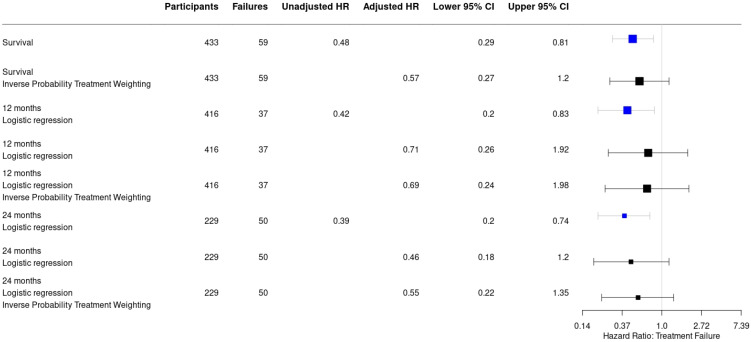
Forest plot displaying a consistent reduction in hazard risk ratios comparing the use of antibiotic-loaded carriers to no antibiotic-loaded carriers, stratified by the analysis performed. Point estimates are given with their 95% confidence intervals. Survival: infection free follow-up; failures: recurrence of infection.

**Figure 4 antibiotics-11-01330-f004:**
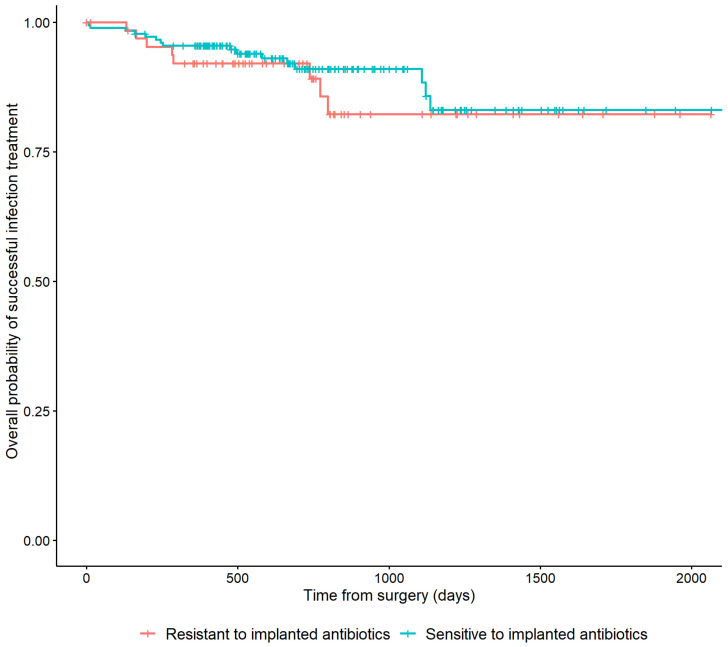
Kaplan–Meier Survivorship curve demonstrating the relationship between the microbiological sensitivity to the local antibiotics that have been applied and recurrence of FRI.

**Table 1 antibiotics-11-01330-t001:** Baseline patient characteristics.

	Cohort (n = 433)	Local Antibiotics Group (n = 251)	No Local Antibiotics Group (n = 182)	*p*-Value
	n (%)	n (%)	n (%)	
Sex (male)	318 (73.4)	191 (76.1)	127 (69.8)	0.142
Age (years): median (P_25_–P_75_)	50.6 (36.3–62.0)	50.0 (37.0–62.0)	51.5 (33.0–63.0)	0.750
BMI (kg/m^2^): median (P_25_–P_75_)	27.0 (23.3–30.4)	27.7 (23.5–31.0)	26.3 (23.1–29.7)	0.068
Smoker (yes)	112 (25.9)	56 (22.3)	56 (30.8)	**0.047 ***
Diabetes (yes)	47 (10.9)	28 |(11.2)	19 (10.4)	0.813
Immunosuppressives	17(3.9)	9 (3.6)	8 (4.4)	0.668
Renal function (eGFR)				0.114
Normal (≥90 mL/min/1.73 m^2^)	310 (71.6)	178 (70.9)	132 (72.5)	
Impaired (<90 mL/min/1.73 m^2^)	115 (26.6)	71 (28.3)	44 (24.2)	
Not determined	8 (1.8)	2 (0.8)	6 (3.3)	
Fracture characteristics (Location)				
Humerus	20 (4.6)	11 (4.4)	9 (4.9)	0.783
Forearm	26 (6.0)	16 (6.4)	10 (5.5)	0.704
Femur	94 (21.7)	65 (25.9)	29 (15.9)	**0.013 ***
Tibia	219 (50.6)	130 (51.8)	89 (48.9)	0.552
Tibia and fibula	6 (1.4)	6 (2.4)	0 (0)	**0.042 ***
Fibula	13 (3.0)	12 (4.8)	1 (0.5)	**0.011 ***
Pelvis	26 (6.0)	4 (1.6)	22 (12.1)	**<0.001 ***
Clavicle	6 (1.4)	0 (0)	6 (3.3)	**0.005 ***
Tibia and talus	1 (0.2)	1 (0.4)	0 (0)	1.000
Calcaneus	11 (2.5)	4 (1.6)	7 (3.8)	0.215
Midfoot	3 (0.7)	1 (0.4)	2 (1.1)	0.575
Foot (crush)	5 (1.2)	0 (0)	5 (2.7)	**0.013 ***
Sternum	1 (0.2)	0 (0)	1 (0.5)	0.420
Patella	2 (0.5)	1 (0.4)	1 (0.5)	1.000
Fracture healed at time of surgery				**<0.001 ***
Yes	190 (43.9)	163 (64.8)	27 (14.8)	
No	243 (56.1)	88 (35.1)	155 (85.2)	
Time to onset of infection (weeks): median (P_25_–P_75_)	44.0 (6.0–356.0)	125.0 (39.0–806.0)	6.5 (3.0–38.0)	**<0.001 ***
Surgical approach				**<0.001 ***
DAIR	139 (32.1)	22 (8.8)	117 (64.3)	**<0.001 ***
Exchange to new internal fixation	24 (5.6)	10 (4.0)	14 (7.7)	0.096
Fixation removed	95 (21.9)	73 (29.1)	22 (12.1)	**<0.001 ***
Exchange to external fixation	68 (15.7)	51 (20.3)	17 (9.3)	**0.002 ***
Internal fixation	1 (0.2)	0 (0)	1 (0.5)	0.420
No fixation used	106 (24.5)	95 (37.8)	11 (6.0)	**<0.001 ***
Soft tissue status				**0.012 ***
Direct closure possible	300	162 (64.5)	138 (75.8)	
Direct closure not possible (SSG, local/free flap)	133	89 (35.5)	44 (24.2)	
Microbiology				
Monomicrobial	199 (46.0)	123 (49.0)	76 (41.8)	0.144
Polymicrobial	154 (35.6)	65 (25.9)	89 (48.9)	**<0.001 ***
Culture negative	80 (18.5)	63 (25.1)	17 (9.3)	**<0.001 ***

P_25_–P_75_: 25th and 75th percentile; DAIR: debridement, antibiotics and implant retention; eGFR: estimated glomerular filtration rate. eGFR was considered impaired when <90 mL/min. SSG: split skin graft; * statistically significant at *p* < 0.05 (in bold).

**Table 2 antibiotics-11-01330-t002:** The effect of ALCs and the susceptibility to implanted antibiotics on recurrence of infection.

Characteristics	n	Recurrence of Infection n (%)		HR	95% CI
		Yes	No		
Antibiotic loaded carrier used (all time points)	433				
Yes	251	25 (10.0)	226 (90.0)	0.48	0.29–0.81 *
No	182	34 (18.7)	148 (81.3)		
Susceptible to local antibiotics	244				
Yes		16 (9.0)	162 (91.0)	0.75	0.32–1.74
No		8 (12.1)	58 (87.9)		
Antibiotic-loaded carrier bioabsorbable	251				
Yes		16 (7.5)	197 (92.5)	-	-
No		9 (23.7)	29 (76.3)		

HR: hazard ratio, 95% CI: 95% confidence interval, * unadjusted survival analysis.

**Table 3 antibiotics-11-01330-t003:** Overview of the antibiotics used in the antibiotic-loaded carriers and the recurrence rate of infection related to the susceptibility to the implanted antibiotics.

Antimicrobial Agents	n (%)	Susceptible to Local Antibiotics		Recurrence of Infection
		No n = 66 (%)	Yes n = 178 (%)	n = 58 (%)
Gentamicin	221 (51.9)	64 (29.0)	157 (71.0)	22 (10.0)
Gentamicin and Vancomycin	11 (2.6)	1 (9.1)	10 (90.9)	1 (9.1)
Gentamicin and Clindamycin	3 (0.7)	1 (33.3)	2 (66.7)	1 (33.3)
Gentamicin and Tobramycin	6 (1.4)	0 (0)	6 (100)	0 (0)
Gentamicin and Vancomycin and Colistin	1 (0.2)	0 (0)	1 (100)	0 (0)
Vancomycin	2 (0.5)	0 (0)	2 (1.1)	0 (0)
None	182 (43.0)	-	-	34 (18.7)

Seven patients with an unknown susceptibility to the local antibiotics were excluded from the analysis. The local antibiotics used in this group were: gentamicin in two cases, gentamycin and tobramycin in two cases, tobramycin in two cases and gentamicin and rifampicin in one case.

**Table 4 antibiotics-11-01330-t004:** Overview of the antibiotic carriers.

Antibiotic Carriers	n (%)
Cerament G	147 (58.6)
Cerament V	1 (0.4)
Herafill G	11 (4.4)
Osteoset T	2 (0.8)
PMMA	38 (15.1)
Cerament G + Herafill	42 (16.7)
Cerament G + Herafill + Cerament V	2 (0.8)
Cerament G + Osteoset T	8 (3.2)

## Data Availability

The data presented in this study are available on request from the corresponding author.

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
