# Peer review of "Does the Use of Local Antibiotics Affect Clinical Outcome of Patients with Fracture-Related Infection?"

_antibiotics, 2022, doi:10.3390/antibiotics11101330_

Round 1
Reviewer 1 Report
In this article, authors report a multinational retrospective study about the use of local antibiotics in fracture related infection.
The paper is well written with results fully analysed through the paper also major biases are to note between ALC and non ALC ( i.e DAIR) that make readers doubt the efficicency of local antibiotcs
I have minor comments regarding the paper :
- put et al in italics as well as bacterial specied in the table .
- no need to put (S.aureus) next to Staphylococcus aureus
- what are the criteria that make surgeons decide for ALC or not ALC? Are attitudes the same in different centers. Please precise this
- Since DAIR is frequently performed in the group non ALC it would be interesting to present and discuss bacterial data of reccurence to check if it corresponds to the same microorganism than the first episode of infection
overall with minor comments taken into consideration, I Believe this work deserves to be published and could enrich the bibliography in this field
Author Response
In this article, authors report a multinational retrospective study about the use of local antibiotics in fracture related infection.
The paper is well written with results fully analysed through the paper also major biases are to note between ALC and non ALC ( i.e DAIR) that make readers doubt the efficicency of local antibiotcs
I have minor comments regarding the paper :
- put et al in italics as well as bacterial specied in the table .
Response: Thank you for pointing this out, et al and bacterial species names were put in italics.
- No need to put (S.aureus) next to Staphylococcus aureus
Response: Thank you for this comment, (S. aureus) has been removed.
- what are the criteria that make surgeons decide for ALC or not ALC? Are attitudes the same in different centers. Please precise this
Response: Thank you for this valid comment. To date there are, to our best knowledge, no evidence-based guidelines on local antibiotics that could be used per clinical scenario. This study retrospectively evaluated the practices at the participating centres. Decisions on whether or not to use ALC and the type of ALC were based on the preferences of the treating physicians. We have made it clear that this is not a randomized trial. This study stresses that there should be more research on improving the knowledge on applying ALCs in different clinical scenarios. All three centres participating in this study are tertiary referral centres experienced in treating bone and joint infections.
- Since DAIR is frequently performed in the group non ALC it would be interesting to present and discuss bacterial data of recurrence to check if it corresponds to the same microorganism than the first episode of infection.
Response: Thank you for this valid comment. We made a new table of pathogens isolated at initial infection and pathogens isolated at the time of recurrence. This table was added to the supplementary materials.
Text added to manuscript: line 106-111: “The pathogens cultured at initial FRI treatment compared to the pathogens cultured at time of recurrent FRI are displayed in supplementary materials (Table S1). Identical pathogen, isolated at the initial FRI treatment and at the time of recurrence, were found in 10/34 (29.4%) of cases in the non-ALC group and in 9/25 (36.0%) of cases in the ALC group.”
overall with minor comments taken into consideration, I Believe this work deserves to be published and could enrich the bibliography in this field
Reviewer 2 Report
I have a few queries while reading your article.
1. Section 2.2-Rationale for using Gentamycin as the most common antimicrobial agent?
2. Section 2.2-Rationale for the combination of antimicrobial agents?
3. Table 3 ;footnote-Why Rifampicin was used in a single case out of all?
4. The final follow-up was done at 24 years. I believe this is debatable as FRI needs long term follow-up study to arrive at a solid conclusion
5. Table 1-Highly heterogeneous cohort including smokers, immunosuppressive patients, diabetics and renal patients is not acceptable in assessing outcomes in FRI.Kindly do a separate subgroup analysis
6.Table 1;Surgical approach-Rationale for opting internal fixation as treatment in FRI ?
7.Table 1;Surgical approach-KIndly provide the protocols for the treatment plan of EXCHANGE TO INTERNAL FIXATION ?
8.Any validated tools used to assess the severity of infection ?
9.How was selection bias eliminated in this study ?
10.Please enumerate the confounding variables that have come across in this stud,steps taken to eliminate this and whether this was done satisfactorily ?
We look forward to hearing from you soon.
Author Response
I have a few queries while reading your article.
- Section 2.2-Rationale for using Gentamycin as the most common antimicrobial agent?
Response: Thank you for this comment. Gentamicin is the most frequently used and widely studied antibiotics used in antibiotic loaded carriers. It is thermally stable and provides broad bactericidal coverage. It is also the most widely available antibiotic in commercially-available, licensed local antibiotic preparations.
Text added to manuscript: line: 121-123: “Gentamicin is one of the most frequently used and widely studied antibiotics in ALC. It is thermally stable and provides broad bactericidal coverage, which makes it a suitable agent for ALC.”
- Section 2.2-Rationale for the combination of antimicrobial agents?
Response: Thank you for this valid question. In some cases, it is thought to be beneficial to cover a broader range of pathogens by combining antimicrobial agents. This was based on the preference of the treating surgeon. It is usually performed when there is previous microbiology which might suggest a pathogen which could be resistant to gentamicin alone.
- Table 3 ;footnote-Why Rifampicin was used in a single case out of all?
Response: Thank you for this question. The use of rifampicin in bone cement is an off-label treatment option. It has not been extensively studied and may alter the biological characteristics of the antibiotic carriers. Rifampicin has shown to be able to infiltrate osteocytes, kill the bacteria that are hiding in the cells and let the osteocyte recover to a healthy cell. However, as this was a retrospective evaluation of the practices in our centres we, unfortunately, cannot track down the rationale of using rifampicin in this single case. It was a single surgeon preference.
- The final follow-up was done at 24 months. I believe this is debatable as FRI needs long term follow-up study to arrive at a solid conclusion
Response: Thank you for this valid comment. We agree that long term follow-up would be superior to evaluate the clinical outcome. However, studies have shown that the large majority of recurrent infections occur within the first year after treatment for the initial infection was finished. Data on follow-up of PJI and osteomyelitis is available but it is much more difficult to be precise about adequate follow-up of FRI cases. In unhealed infected fractures, recurrence almost always (>90%) occurs in the first year (1).
- McNally, Martin & Ferguson, Jamie & Dudareva, Maria & Palmer, Antony & Bose, Depa & Stubbs, David. (2018). For how long should we review patients after treatment of chronic osteomyelitis? An analysis of recurrence patterns in 759 patients. Bone and Joint Journal Orthopaedic Proceedings Supplement 2017;99 (Supp 22):22.
- Table 1-Highly heterogeneous cohort including smokers, immunosuppressive patients, diabetics and renal patients is not acceptable in assessing outcomes in FRI. Kindly do a separate subgroup analysis
Response: Thank you for pointing this out. We agree that there is heterogeneity in our cohort, however, this cohort reflects clinical daily practice. We corrected for confounding by calculating the chances that ALC would be used in a patient based on baseline characteristics using inverse probability for treatment weighting. The effect of ALC on outcome was evaluated by using the IPTW propensity score in a multivariable model. We have shown in another study on this cohort that smoking was a factor in outcome and we have added this data here. However, multiple subgroup analyses of smaller subgroups would reduce the power of the analysis to a point where any conclusions would not be valid. For example, we only had 47 diabetic patients. We are therefore reluctant to do this, as it may give a false impression of the contribution of these factors. IPTW scoring is designed to deal with this difficulty in the statistical analysis.
- Table 1;Surgical approach-Rationale for opting internal fixation as treatment in FRI?
Response: Exchange to new internal fixation was performed when there was a need for fracture stabilization in case fracture healing had not occurred at time of infection treatment. The choice to perform a single- or two-stage exchange was based on the preference of the treating surgeon.
Text added to manuscript: line 320-325: “In case the fracture was not healed and stability was required, either an external fixator was used or new internal fixation was performed in a (single) staged fashion to facilitate fracture healing. Generally, internal fixation was only chosen in cases where external fixation would be difficult and when the soft tissue were good or could be improved with soft tissue reconstruction.”
- Table 1;Surgical approach-KIndly provide the protocols for the treatment plan of exchange to internal fixation? Response: Thank you for this comment. Please see point 6. We do not have an a priori protocol for this as it was determined individually by the treating surgeon.
- Any validated tools used to assess the severity of infection?
Response: Thank you for this valid question. Assessing severity of infection is difficult and somewhat subjective. There is no validated outcome measure or scoring system for severity of FRI. It is based on a combination factors (e.g. the fracture healing status, quality of soft tissues covering the infected area, the resistance pattern of the infection causing pathogens, the extent of dead bone that needs to be removed and overall health status of the host). We used the validated BACH criteria to evaluate the complexity/severity of the infection. This was developed to classify osteomyelitis but is not currently advocated in FRI. However, there can still be unmeasured factors that could influence the severity of infection (e.g. intracellular hiding bacteria).For this reason, we have not used it in our analysis. It should be noted that many of the factors which we have included (fracture healing status, requirement for soft-tissue cover, host health status, pathogens etc) are markers of severity and have been included in the analysis.
Text added to manuscript: line: 291-294: “We based severity of infection on the BACH classification for osteomyelitis, but this has not been validated for FRI. However, it is likely that there remain unmeasured factors that could influence the infection severity (e.g. intracellular hiding bacteria).”
- How was selection bias eliminated in this study?
Response: Thank you for this valid comment. We addressed the selection bias by including all consecutive patients with an FRI. All eligible patients were evaluated as described in the study. The chance of being ‘selected’/ treated with ALC was accounted for by performing the IPTW analysis as explained in more detail in point 10.
- Please enumerate the confounding variables that have come across in this stud,steps taken to eliminate this and whether this was done satisfactorily ?
Response: Thank you for addressing this important principle. We approached this problem by designing a study which would include all types of cases which commonly are referred for treatment. We then accurately recorded a large number of potential confounding factors and employed a modern analysis technique (IPTW) which allowed weighting of the effect of each factor on possible outcome (positive or negative). Advice was taken from the Oxford Centre for Evidence-based Medicine in the design. To be able to say something about causality of how the ALC affect the outcome of FRI, we made a directed acyclic graph (DAG) prior to analyzing our data. DAGs can help to visualize factors (confounders, colliders and mediators) that distort the causal relationship between the use of ALC and outcome of patients with FRI. By performing inverse probability for treatment weighting, we addressed the heterogeneity in our cohort to estimate the chance that a patient would be treated with ALC based on baseline patient characteristics. This resulted in a propensity score which was used in our multivariable logistic regression model. The model adjusted for confounders we determined in the DAG. Although there will always be unmeasured/unmeasurable confounding factors, we believe our approach to limit confounding was the best achievable.
Thank you for your constructive comments. We hope that we have been able to address these. We believe the amendments have improved the paper and look forward to hearing from you soon.
